# Measuring Structural Changes in Cytochrome *c* under Crowded Conditions Using In Vitro and In Silico Approaches

**DOI:** 10.3390/polym14224808

**Published:** 2022-11-09

**Authors:** Zahoor Ahmad Parray, Ahmad Abu Turab Naqvi, Ishfaq Ahmad Ahanger, Mohammad Shahid, Faizan Ahmad, Md. Imtaiyaz Hassan, Asimul Islam

**Affiliations:** 1Centre for Interdisciplinary Research in Basic Sciences, Jamia Millia Islamia, Jamia Nagar, New Delhi 110025, India; 2Department of Chemistry, Indian Institute of Technology Delhi, IIT Campus, Hauz Khas, New Delhi 110016, India; 3Department of Chemistry, Biochemistry and Forensic Science, Amity School of Applied Sciences, Amity University Haryana, Gurugram 122413, India; 4Department of Basic Medical Sciences, College of Medicine, Prince Sattam Bin Abdulaziz University, Al Kharj 11942, Saudi Arabia; 5Department of Biochemistry, School of Chemical and Life Sciences, Jamia Hamdard, New Delhi 110062, India

**Keywords:** cellular crowding, cytochrome *c*, synthetic crowders, circular dichroism, molecular docking

## Abstract

It is known from in vitro studies that macromolecular crowding in the cell effects protein structure, stability and function; but predictive studies are relatively unexplored. There are few reports where the effect of various crowder mixtures has been exploited to discern their combined effect on the structural stability of proteins. These studies are more significant because their effect can mimicked with in vivo conditions, where the environment is heterogeneous. Effects of two crowders, polyethylene glycol (PEG 400 Da), and its monomer ethylene glycol (EG) alone and in mixture on the structural stability of cytochrome *c* (cyt *c*) were determined using various spectroscopic and bioinformatics tools. The main conclusions of our study are (i) the monomer EG has a kosmotropic effect on the protein (stabilizes the protein), and has no significant effect on the tertiary structure; (ii) PEG 400 destabilizes the structure as well as the stability of the protein; and (iii) EG counteracts the destabilizing effect of PEG 400. From this investigation, it seems evident that proteins may fold or unfold in the crowded environment of the cell where various interactions assist them to maintain their structure for their functions. Bioinformatics approaches were also used to support all of the in vitro observations. Cyt *c* is functional protein; if the structure of the protein is modulated due to change in the environment its nature of function will also change. Our research addresses the question by modulating the environment around the protein, and the macromolecule (protein) conformation dynamics and interaction study via in vitro and in silico approaches which indirectly compares with that of the environment in-cellular milieu, which is highly crowded.

## 1. Introduction

The macromolecules such as proteins, carbohydrates, nucleic acids, and ribosomes are present in a large number, and have evolved to function in a crowded media [1]. Since the interior of a cell is always occupied by macromolecules, it is interesting to ask how proteins fold and function in such environment. This is particularly important because information regarding structure, folding and function of the test protein are usually obtained from in vitro studies in dilute solutions; but the environment of the test protein inside the cell and that in the dilute buffer (idealized conditions) are totally different from each other [2,3,4]. The cellular environment is heterogeneous, complex, and crowded [1], on the contrary, in vitro environment is homogeneous that is employed in almost all biophysical studies. Thus to understand the effect of macromolecular crowding on structure, folding and function of the test protein, natural and synthetic crowders are used to mimic the cell environment in the in vitro experiments [5].

In biological systems, various interactions maintain stability and structure of different macromolecules in a highly crowded milieu, which include hydrogen bonding and ionic, van der Waals and hydrophobic interactions. Biopolymer shape and function in biological cells are controlled by depletion forces and macromolecular crowding [6,7,8,9,10,11,12,13,14,15]. Depletion force is observed in crowded conditions such as those in the living cell that contains soluble proteins and other molecules (micro and macromolecules) which constitute 20–30% of the cell volume [7,12,16]. The depletion attraction is driven by an increase in the volume available to a macromolecular solute, although the hydrophobic effect is driven by an increase in the hydrogen-bonding states available to water [17]. They are also superficially similar in that one is solely controlled by entropic effects, while the other is primarily driven by entropic effects [17]. The bio-macromolecular interactions and activities in a crowded environment of in vitro (dilute solutions) and cellular (in vivo) conditions show wide range of substantial differences [13]. For protein stability and folding, the protein-protein or protein-crowder interactions in a system is very important and such interactions leads either stabilization or destabilization of proteins [18,19]. These interactions are divided into preferential interactions (steric repulsions or hard interactions), which depends on either excluded volume effect or/and soft interactions also called chemical interactions [20,21]. Around one-third of the macromolecule’s accessible volume is excluded in a concentrated solution because of the crowded milieu [13]. This is interpreted as the combined outcome of both general types of interactions namely, (i) excluded volume, which is the result of inaccessibility of molecules to occupy space because of the presence of background macromolecules, and (ii) chemical effects (soft interaction), caused by the attraction and repulsion between molecules [20,21]. The proposed assumption that repulsive chemical interactions increase the excluded volume is simple to understand. If the test protein and crowder molecule have the same charge, the resulting repulsion increases. As a result, repulsive interactions increase in excluded volume, with the greater size of the crowder molecule playing a key role. Soft interactions can be repellent or attractive, and therefore improve or disrupt the structure of the test protein [20,22]. As a result, unlike hardcore repulsions which often stabilize proteins, chemical (soft) interactions either destabilize or stabilize proteins [23].

The mixed crowding induces greater cooperativity in the domain movement as compared to the components of the mixtures confirmed by various studies in recent times [18,23,24,25]. When compared with those obtained for individual crowders, experimental investigations had shown considerable deviations from ideality, with an ideal solution being measured to be taken from the sum of the contributions of those achieved in the presence of crowder molecules alone [18,24,25,26]. Also, the investigators suggested out of their experiments and analysis that the small size of the crowder in the mixture is usually the leading factor in the stabilization of proteins [18,24,25]. PEG 400 is strongly hydrophilic, soluble in water, acetone, alcohols, benzene, glycerin, glycols, and aromatic hydrocarbons, and is slightly soluble in aliphatic hydrocarbons. It is a clear, colorless, viscous liquid. Due to its low toxicity, PEG 400 is widely used in a variety of pharmaceutical formulations [27,28]. The viscosity of PEG is 90 cST (centistokes) with low melting point (4–8 °C), (https://en.wikipedia.org/wiki/PEG_400, accessed on 17 June 2022). PEG 400 has been seen improving drug loading capacity and had been reported in prolongation of the survival of rats suffering malignant brain tumors [28]. PEG 400 had been shown to interact with various proteins via hydrogen bonds in dilute solutions, such interactions between PEG-protein leads formation of either intermediate states or leads protein destabilization [22,29,30,31,32,33]. Ethylene glycol (EG) is small molecule, thick, odorless, colorless, sweet-tasting, toxic and viscous liquid. The melting temperature of EG is very low (−12.9 °C) and boiling temperature is very high (197.3 °C). Ethylene glycol is synthesized from ethane via intermediate ethane oxide, also is produced when ethylene oxide and water reacts. It is primarily utilized for two things: as a raw material to make polyester fibers and as an ingredient in antifreeze compositions. When combined with water, ethylene glycol freezes at a lower temperature, normally at −12 °C. Ethylene glycol is usually applied as a preservative for biological specimens as a preferable alternative to formaldehyde [34]. Water and ethylene glycol (EG) are completely miscible, and EG has a significantly higher viscosity than pure water. As a result, EG-water mixtures are a notable example of viscosity-controllable media. This makes them valuable for studying the impacts of solvent dynamics on electron transfer reactions [35], which are sometimes the dominant contributors to the reaction rate. At the molecular level, the structural and dynamical aspects of water/EG combinations are far more intricate than those of pure components. The features of the hydrogen bond network in the mixtures are frequently used in the interpretation [35,36].

Recently, we have reported that PEG 400 and EG individually have opposite effects on the tertiary structure of cyt *c* [34,37], high concentration of PEG 400 (300 mg/mL) induced molten globule [32], and EG showed stabilizing effects on the protein due to protein-solvent preferential interaction or/and kosmotropic effect [37]. Myoglobin (Mb) retained its structure and was stabilized in the presence of mixture of crowders (PEG 400 and EG) [18]. These studies [18,22,25,32,37] and the present study of proteins in cellular-like conditions provide insights towards the importance of various interactions in the living cell environment [38]. However, most of such studies were carried out in the presence of individual crowders and don’t perfectly mimic the conditions of the cell. Though, here we used two different sizes of molecules, both individually and in mixture, to compare the changes in the protein in crowded environment. This study showed that crowders alone and crowders in the mixture behave differently with the protein (cyt *c*). Therefore, this field of research called macromolecular crowding is now emerging as one of the most thriving in molecular and cell biology. Moreover, molecular dynamics (MD) simulations for this protein (cyt *c*) in the presence of these crowders (both individually and in mixture) were not carried out before, which is important to study the protein dynamics at atomic and molecular levels [39].

MD simulation studies support the in vitro studies to understand the changes at the atomic levels. The atomic level details are therefore helpful to understand the functional aspects of the biomolecule as a result of structural changes. The computational methods (molecular docking and MD simulations) carried out within conditions closely reflect the studies carried out in experiments (in vitro), suggested that protein folding or unfolding under in vitro system may be exploited and defined by computational simulations [39].

In this study effects of PEG and EG alone and in combination at PEG:EG concentration ratios 1:1, 1:6 and 6:1 on the structure of cytochrome *c* (cyt *c*) were carried out using in vitro and in silico approaches. The bioinformatic approaches support the data obtained by biophysical techniques which measures the changes in the structure of the protein (cyt *c*) under crowded systems (PEG 400 + EG). The experimental outcomes and the theoretical predictions provide strong evidence how crowders (individually and in the mixture) influence the protein at atomic levels.

## 2. Materials and Methods

### 2.1. Materials

Commercially available lyophilized form of horse heart cytochrome *c* (cyt *c*) was procured from Sigma chemical company (St. Louis, MO, USA. Polyethylene glycol (PEG 400 Da) and ethylene glycol (EG) were procured from Merck (Maharashtra, India). Reagents (mono- and dibasic sodium phosphate salts) used to prepare phosphate buffer were purchased from Himedia (Einhausen, Germany). The filters of pore size equal to 0.22 μm were purchased from Merck Millipore Ltd. (Tullagreen, Cork, Ireland).

### 2.2. Methods

#### 2.2.1. Preparation of Protein and Reagents

As described earlier [32,37], 70 mg/mL of lyophilized powdered form of cyt *c* was dissolved in 50 mM phosphate buffer and oxidized by potassium ferricyanide [40]. To remove excess of potassium ferricyanide, the protein solution was dialyzed against 2 L of phosphate buffer (pH 7.0) solution at 4 °C with several changes. The dialyzed protein solution was filtered using a 0.22 μm Millipore filters. The equation (*c* = *A*_410_/*εl*) was applied to determine the protein stock concentration, where *c*, *l*, *A*_410_ and *ε* are concentration in a molar, path length of cuvette in centimeter (cm), absorbance at the wavelength of 410 nm, and molar absorption coefficient at 410 nm (*ε*_410_, M^−1^ cm^−1^), respectively. The value of *ε*_410_ was 106,100 M^−1^ cm^−1^ [41].

The crowder molecules (PEG 400 and EG) and denaturant (guanidinium chloride, GdmCl) were diluted in the phosphate buffer before filtration with Whatman filter paper No. 1. The stock solution concentrations were determined using refractive index readings as per reports, for crowders [42,43,44] and GdmCl [45]. To conduct optical measurements, the appropriate solutions were prepared in degassed buffer solution. Each sample of varied crowder concentrations (individual and in combinations) was made in triplicate and incubated overnight at 25 ± 1 °C.

#### 2.2.2. Spectroscopic Techniques

##### UV-Visible Spectra Measurements

For spectral measurements, a Jasco V-660 UV-vis spectrophotometer connected to a Peltier temperature regulator (ETCS761) was used. All measurements of absorbance were conducted using 1.0 cm path length cuvette in the region of 240–700 nm (near-UV and Soret- absorption), and 6 to 7 µM of the protein was used in each sample [34,37].

##### Circular Dichroism (CD) Measurements

A Jasco Spectropolarimeter (J-1500 model) with a circulation bath (MCB-100) was utilized to perform circular dichroism (CD) spectral measurements to know the effect on the secondary and tertiary structure of the protein in the absence and presence of crowders. The CD spectra measurements were carried out in the near- and far-UV regions using a protein concentration of 16 µM for each sample in 1.0 and 0.1 cm path length cuvette, respectively. Moreover, a 1.0 cm path length cuvette was also used in the measurements of Soret CD in the range of 370–450 nm. 4–5L min^−1^ of nitrogen was flushed constantly to lessen the level of noise. More than 3 accumulations for each sample with baseline were run for the improvement of better signal/noise. The raw CD signal at a wavelength λ (nm), *θ*_λ_ (mdeg) was converted to concentration-independent parameters i.e., [*θ*]_λ_ (deg cm^2^ dmol^−1^) the mean residue ellipticity (MRE) using the following equation:[*θ*]_λ_ = M_0_*θ*
_λ_/10*lc*(1)
where M_0_ is the protein mean residue weight, *c* is the protein concentration in gm/cm^3^, and *l* is the path length of the cuvette in centimeters. [*θ*]_222_ and [*θ*]_208_ probes were used to check changes in secondary structure and estimation of α-helical content of the protein in various crowded systems as described earlier [18,46].

##### Fluorescence Spectra Measurements

Fluorescence measurements were performed using a Jasco FP-6200/STR-312 spectrofluorimeter. The emission and excitation slits were set to 10 nm, and 1.0 cm path length cuvette was used for all measurements. The excitation wavelength employed for measurements was 280 nm, while the wavelength range addressed for emission spectra was 300–400 nm as described earlier [32,37].

#### 2.2.3. Computational Studies

##### Molecular Dynamic (MD) Simulations

MD simulations are useful for studying atomic motions of the molecular systems [46,47]. This enables them to complement the experimental studies as they do not look deep at the atomic level incidents. The atomic level details are therefore helpful to understand the functional aspects of the biomolecule, as a result of structural changes. We performed simulations to study atomic motions and inter-atomic interactions in cyt *c* that define its structure and function. The GROMACS 2021 [47] package was used to perform MD simulations on the protein in water and in solutions of PEG 400, EG, mixture of PEG and EG and GdmCl. The protein databank, PDB (www.rcsb.org/pdb, accessed on 20 November 2021), was used to get three-dimensional coordinates of horse cyt *c* (PDB ID: 1HRC) [48]. Using the Swiss-PDB viewer, the structure was examined and evaluated for missing atoms and side chain repairs. All the solvent molecules’ force-field parameters were produced using the CGenFFserver (https://cgenff.umaryland.edu/ (accessed on 20 November 2021)) which provides CHARMM (chemistry at Harvard macromolecular mechanics) force field parameters for molecules. The CGenff charmm2gmx script produced by Alex MacKarrel Lab (http://mackerell.umaryland.edu/charmmff.shtml, accessed on 20 November 2021)), was used to convert the generated parameter files to GROMACS format. For atomic parameterization, the CHARMM36 force field was employed which contains force field parameters for heme prosthetic group also (cyt *c* also contains heme). All the systems (the protein with and without crowder and denaturant) were solvated using the SPCE water system that came with the GROMACS suit prior to MD simulations. Chlorine ions were then introduced to the system to neutralize the charge. The Steepest Descent Algorithm integrated with GROMACS suit was used to minimize the energy of all the systems, with a step size of 0.01 nm and a tolerance of 1000 kJ/mol/nm. After minimization, the systems were equilibrated for a 100 ps period under NVT (constant volume) and NPT (constant pressure) ensemble conditions. Eventually, all the properly minimized and equilibrated systems were subjected to 100 ns MD simulations each. MD simulation trajectories were analyzed and structural properties such as radius of gyration (*R*_g_), root mean squared deviation (RMSD), root mean squared fluctuation (RMSF), solvent accessible surface area (SASA) and secondary structural elements were calculated to see the effects of solvents on cyt *c* conformation.

##### Molecular Docking

PyRx software was used to dock EG and PEG 400 to a macromolecule (cyt *c*) to explore compounds with specified biological functions for virtual molecular screening [49]. PyRx code is designed in Python and has an easy-to-use interface that runs on all major operating systems (Windows, Linux, and Mac OS). It is a mash-up of numerous software packages including AutoDockVina, Open Babel, Mayavi, and others. PyRx’s docking software packages are Vina and AutoDock 4.2 [50]. Using Autodock software, the input files ligand, EG and PEG 400 (Source PubChem), and macromolecule, cyt *c* (PDB id: 1hrc) in .pdb format were converted to .pdbqt files. Following the preparation of the content files, docking was conducted using AutoDock 4.2 and Vina. Grid cell dimensions were adjusted to 50, 42, and 55 for the X, Y, and Z conformations, respectively. The grid area was correctly specified, allowing the receptor to pick the search space for docking with the ligand properly at the binding site. The Lamarckian Genetic Algorithm (LGA) was used to assess the interaction between cyt *c* and the respective crowders (EG and PEG 400). Following the completion of the Vina calculations, the software provided results of binding affinity energy (kcal/mol) of possible conformations of the macromolecule with both crowders in the form of a table. Finally, the best-docked protein-crowder molecule complexes (EG-cyt *c*-PEG 400) were modified and analyzed using PyMOL [51], BOVIA Discovery Studio visualization to build 2D interaction plots [52,53], and LigPlot v.4.5.3 offered by EMBL-EBI, which employs Java as a programming language.

## 3. Results

### 3.1. Measuring Structural Changes in Cyt c Caused by the Mixture of Crowders (PEG 400 + EG)

#### 3.1.1. Absorption Spectroscopy

Figure 1A shows the effect of different concentrations (mg/mL) of EG and PEG 400 alone and their mixtures on the absorption bands of cyt *c* in near-UV and Soret regions. This figure also shows spectrum of the protein in the presence of 6 M GdmCl as a reference spectrum of the unfolded cyt *c*. The inset of the figure depicts plots of absorption coefficients at 409 and 280 nm (*ε*_409_ and *ε*_280_) versus various concentration ratios of PEG 400 and EG, [PEG 400 + EG]. This figure suggests that the values of *ε*_409_ and *ε*_280_ of cyt *c* increase in the presence of EG, decrease in the presence of PEG 400 and in the presence of mixture of PEG 400:EG (1:1 and 1:6) there is no significant increase or decrease in the absorbance of cyt *c*, in comparison of native state, the mixture of PEG 400:EG (6:1) decreases the absorbance significantly, yet the change was smaller than PEG 400 alone.

#### 3.1.2. Tryptophan Fluorescence Measurements

Figure 1B shows the effect of various mixture ratios of crowders (PEG 400 and EG) on the fluorescence emission spectra of cyt *c*. The inset of the figure depicts a plot of fluorescence emission values (black circles) at 342 nm, *F*_342_ versus [PEG 400 + EG]. This figure shows that (i) PEG 400 increases the fluorescence emission in cyt *c*, (ii) there is a decrease in Trp fluorescence in the presence EG at concentrations of 50 and 300 mg/mL, (iii) in the presence of mixtures of PEG 400:EG (1:1 and 1:6) there is decrease in the fluorescence emission of cyt *c*, and (iv) the mixture of PEG 400:EG (6:1) increases the fluorescence of cyt *c* significantly, yet the change was smaller than that of PEG 400 alone.

#### 3.1.3. Near-UV and Soret Circular Dichroism (CD) Measurements

Figure 1C depicts the effect of different mixture ratios of crowders (PEG 400 and EG) on the near-UV CD spectra of cyt *c*. It is known that cyt *c* (in buffer) displays two strong negative bands around 282 and 289 nm [54,55]. The inset in Figure 1C shows plots of mean residual ellipticity at 282 and 289 nm ([*θ*_282_] and [*θ*_289_]) versus [PEG 400 + EG]. This figure depicts that near-UV CD signals around 282 and 289 nm of cyt *c* (i) increase by small change due to EG alone, (ii) decrease significantly by PEG 400 alone, (iii) increase in the presence of mixture of PEG 400:EG (1:1 and 1:6), and (iv) the mixture of PEG 400:EG (6:1) decreases the CD signals of cyt *c* significantly, yet the change was smaller than that of PEG 400 alone. This figure also shows that the near-UV CD spectra of cyt *c* completely disappeared in the presence of GdmCl (6 M).

Figure 1D shows the Soret CD spectra of cyt *c* in the absence and presence of different concentration ratios of crowders [PEG 400 and EG] and high concentration of GdmCl (6 M). The inset of the figure shows plots of mean residual ellipticity at 405 and 416 nm ([*θ*]_405_ and [*θ*]_416_) versus [PEG 400 + EG]. It is seen in this figure that EG alone shows significant increase in CD value at wavelength of 405 nm, [*θ*]_405_, however CD value at wavelength of 416 nm, [*θ*]_416_ is not affected. But PEG 400 alone affect the values of both [*θ*]_405_ and [*θ*]_416_, hence heme-globular interaction and heme-Met80 and Phe82-Fe interactions, respectively. PEG 400:EG mixture ratios (1:1 and 1:6) show insignificant change in the CD signals of cyt *c* around 405 nm and significant increase around 416 nm when compared with that of the native protein. Mixture ratio 6:1 (PEG 400:EG) increases the CD signals of cyt *c* insignificantly around 405 nm with a shift and small decrease in the CD signal can be seen around 416 nm (negative peak), yet the change is smaller than that due to PEG 400 alone. This figure also shows that CD signal of cyt *c* around 405 nm ([*θ*]_405_) which increases significantly and signal around 416 nm ([*θ*]_416_) completely disappeared in the presence of GdmCl (6 M).

#### 3.1.4. Far-UV Circular Dichroism (CD) Measurements

Figure 2A shows far-UV CD spectra of cyt *c* under different solvent conditions. Two negative bands at 208 and 222 nm are characteristics of a typical α-helical protein. This figure shows that these characteristics are unaffected by PEG 400, a significant increase occurs in the presence of EG, and in the presence of mixture ratios, PEG 400:EG (1:1, 1:6 and 6:1) there is an increase the CD signals of cyt *c*. The inset of the figure displays a plot of [*θ*]_222_ versus [PEG 400 + EG], the different colors in the inset show triplicate data. This figure also shows GdmCl (6 M)-induced denatured CD spectra of cyt *c*.

Figure 2B shows a plot of % α-helical content of cyt *c* in the absence and presence of mixture of crowders at ratios, where black circles represents α-helical content values at 222 nm. It can be seen here that EG alone shows a small increase in the α-helical content of the protein, PEG 400 alone and the mixture of both crowders has no significant effect on α-helical content of the protein (see also Table 1).

### 3.2. Computational Studies (In Silico Approaches)

In the structure-based drug design and protein function prediction system, the protein-ligand interaction is quite a fascinating topic. Molecular docking and molecular dynamics simulations are frequently used together to determine binding affinities, various types of interactions, and the structural stability of various proteins interacted with different ligands.

#### 3.2.1. Molecular Dynamic (MD) Simulation Studies

Parallel to standard experimental studies, computational simulations were used. MD simulations enables in the investigation of atomic dynamics in molecular systems [56,57,58] and commonly used in the theoretical analysis of proteins and other macromolecules, as well as the modifications that happen as a consequence of numerous molecular and physiological processes, mutations, ligand binding, temperature or pH change, chemical-induced denaturation, and molecular crowding [57,59,60,61,62,63]. To study the effects of PEG 400 and EG alone and their mixtures (PEG 400 + EG) and 6 M GdmCl on the conformation of cyt *c*, MD simulations of the protein in the native form (in water), 300 mg/mL (0.75 M) PEG 400, 50 and 300 mg/mL (0.8 M and 4.8 M) EG, two mixtures of PEG 400 and EG [PEG 400 + EG]: 50+ 300 mg/mL (0.8 M + 4.8 M), and 50 + 300 mg/mL (0.8 M + 0.75 M), and GdmCl (6M) were completed.

Panels A, B, C, and D of Figure 3, respectively, show plots of radius of gyration (*R*_g_), root mean square deviation (RMSD), root mean square fluctuation (RMSF), and solvent accessible surface area (SASA) of cyt *c* in various conditions; water (black), 50 and 300 mg/mL (0.8 and 4.8 M) of EG (Red and green), 300 mg/mL (0.75 M) PEG 400 (blue), two PEG 400 + EG mixtures, 50 + 300 mg/mL (yellow) and 50 + 300 mg/mL (brown), and 6 M GdmCl (grey). *R*_g_ measurements provide information on the opening of protein structure as it indicates the axial scattering of the atoms of the center of mass of the protein [64]. Table 2 provides the change in the values of *R*_g_, RMSD, RMSF, and SASA of the protein in the absence and presence of EG, PEG 400, mixtures of (PEG 400 + EG) and 6 M GdmCl.

It is observed that the values of *R*_g_, RMSD, RMSF and SASA of the protein shows insignificant change in the presence of EG (50 and 300 mg/mL), a significant decrease in the presence of PEG 400 (300 mg/mL), an increase in the mixture of crowders (PEG 400 and EG) and maximum change in 6 M GdmCl (Table 2). It is seen in Figure 3 that for the native conformation (black) the parameter values which goes up around 40 nanoseconds (ns) simulation time, reaches convergence. Similarly, the protein in 6 M of GdmCl solution also shows a gradual elevation in the parameters which escalate untilthe end of the simulation time. In the presence of EG, there is no significant change in any of the parameters (i.e., they are comparable to the native state), however fluctuation is more in the presence of PEG 400 alone. Moreover, the mixture of the crowders (PEG 400 + EG) cause initial decrease in the values of parameters (*R*_g_, SASA, RMSD and RMSF), however these parameters increases in the presence of EG above the concentration of 50 mg/mL and in the presence of EG (50 mg/mL) and PEG 400 (300 mg/mL), comparable to the protein in the presence of PEG 400 alone and with that of the native conformation at end of simulation. From Figure 3 and Table 2 it can be confirmed that EG alone has insignificant effect on the protein, PEG 400 alone and 6 M GdmCl leads protein structure perturbation, and mixtures of crowders (PEG 400 and EG) leading the protein towards a stabilized condition.

For the secondary structural analysis, changes in the ratio of secondary structural components in response to the solvent effects can be seen in Figure 4; (Figure 4A) the native state in water, (Figure 4B,C) EG with 50 and 300 mg/mL (0.8 and 4.8 M), respectively, (Figure 4D) 300 mg/mL (0.75 M) of PEG 400, (Figure 4E) 300 + 50 mg/mL (0.75 M + 0.8 M) mixture of PEG 400 + EG, (Figure 4F) 50 + 300 mg/mL (0.125 M + 4.8 M) mixture of PEG 400 + EG, and (Figure 4G) 6 M GdmCl. The percentage of secondary structure elements of the protein and protein under co-solute conditions (individual and mixture) were not changed. Figure 4 shows the secondary structure of the protein is neither affected by PEG 400 nor EG alone nor in the presence of mixture of crowders. Moreover, the MD simulation of protein in the presence of 6 M GdmCl (Figure 4G) shows a depth at initial time (15–30 s), however the secondary structure was not affected by GdmCl either to that extent (see Figure 4).

#### 3.2.2. Molecular Docking Studies

Figure 5 displays the interaction of PEG 400 and EG with cyt *c* via non-covalent forces without interacting with the heme. Figure 5A provides a cartoon model of the protein (gray) and ball and stick pattern of PEG 400 (blue) and EG (yellow) and shows ligand interactions with various amino acid residues of the receptor. Figure 5B provides the bond distances generated between ligands (PEG 400 and EG) and the protein, in which PEG 400 interacts Arg38, His33 and Gly24 via single bonds of distance 3.26, 2.95, and 3.29 Å, respectively. Asn31 interacted via more than two bonds with distances of 2.81, 3.01 and 2.93 Å. EG interacts with His33 via two bonds with distances of 3.01 and 2.98 Å, and Asn31 via a single bond of distance 2.98 Å. Figure 5C displays the surface image of the protein (cyt *c*) with binding pocket sites for PEG 400 and EG. PEG 400 and EG interact with the amino acid residues of the protein via non-covalent interactions (conventional H-bonding, vander Waals forces, unfavorable accepter-accepter, and unfavorable donor-donor bonds) as seen in the 2D-structure of the model in Figure 5D. Due to the binding of PEG 400 and EG with cyt *c*, the computational study gives binding energies of −3.3 and −2.7 kcal mol^−1^, respectively.

Figure 6 shows interactions (hydrogen bonding, hydrophobic, etc.) between various residues of cyt *c* with both PEG 400 and EG were further demonstrated using LigPlot (a programmed software that automatically generates schematic diagrams of protein-ligand interactions for a given PDB file).

## 4. Discussions

Proteins show various characteristic bands in their spectra such as absorption bands around 280 nm (due to the aromatic side chains of Tyr and Trp residues), the Soret band at 409–410 nm (due to the heme group), and oxy-deoxy tiny bands around the wavelength range of 500–600 nm provide information about the state of iron atoms (oxidized or reduced form) [65,66]. Figure 1 showed absorption spectra of the native cyt *c* (black) with a sharp Soret band around 409 nm [67]. The heme group placed inside the hydrophobic pocket of the folded protein and π-π* electron transitions occurring in the prosthetic group, resulting in an increase inthe peak in the Soret-region. The Soret band position and shape are solely dependent on the geometric position of their own atom with the heme [68,69]. A decrease in the intensity of the band (Soret region) without any shift in the wavelength in the presence of PEG 400 is greatest at 300 mg/mL, which reflects that protein structure is perturbed and heme environment changes towards the polar environment [70,71,72]. This observation is in agreement with that reported previously [32]. EG alone has an insignificant increase in the absorbance at its both low and high concentrations. However, mixtures PEG 400 and EG caused variable changes in the heme environment. For example, a 1:1 ratio (PEG 400:EG) leads to an increase in absorbance without any shift in the wavelength, a concentration ratio of crowders 1:6 (PEG 400:EG) resulted in an increase in the absorbance with a blue shift, and 6:1 (PEG 400:EG) resulted decrease in the absorbance, but less than that of PEG 400 alone, directs protein towards the native state. There was no change in the oxy-deoxy band (500–600 nm) in the presence of crowders alone or their mixtures, indicating that the protein was oxidized under all these conditions. Moreover, change in Trp and Tyr environments monitored by *ε*_280_, occurred from non-polar to polar environment in the presence of PEG 400 alone showed decrease in absorbance, which is comparable to cyt *c* denatured spectrum (6 M GdmCl-induced state). However, in the presence of mixtures (PEG 400 + EG) absorbance (*ε*_280_) increases which is contrary to that of the denatured cyt *c* (Figure 1A). Figure 1B showed an increase in the fluorescence intensity of the protein in the presence of PEG 400 alone, which showed that the energy transfer occurs from Trp to heme results in a change in the distance between heme and Trp. Quenched Trp changes its non-polar environment towards polar, resulting in an increase in the intensity [32]. On the contrary, the fluorescence intensity of the protein decreases in the presence of EG, which in earlier report was confirmed that EG acts as a quencher as the concentration is increased [37]. EG and cyt *c* showed weak interaction and it has been observed that EG acts as kosmotroph (which maintains the conformation of the protein) [37]. However, the mixture of these crowders at all ratios resulted in a decrease in the fluorescence intensity as compared to PEG 400 alone. Furthermore, in support of the absorbance and fluorescence studies, near-UV and Soret CD measurements (Figure 1C,D) showed that PEG 400 has a significant effect on the environment of aromatic residues (Met80 and Phe82) and heme interactions with protein globule. EG has no significant effect on such intra-molecular interactions (heme with globule, Met80, and Phe82), and mixture (PEG 400 + EG) caused a perturbation of the tertiary structure, but was less than that of PEG 400 alone.

In summary, results shown in Figure 1A–D confirm that (i) PEG 400 alone at high concentration perturbs significant amount of the tertiary structure of cyt *c*, (ii) EG alone maintains the conformation of the protein, (iii) (PEG 400 + EG) mixture ratio (6:1) perturb the tertiary structure of the protein, however other mixture ratios (1:1 and 1:6) have no significant effect directs the protein towards stabilization (decrease in the perturbations). It is interesting to recall a study published recently that showed myoglobin (Mb) is more stable in the presence of a various mixtures of crowders (PEG 400 and EG) [18].

To monitor the secondary structure of cyt *c* using various crowding conditions (PEG 400 + EG), the far-UV CD was exploited. This technique is an insightful probe to measure the secondary structural changes and to determine the helical content of the proteins [73]. The far-UV CD spectrum of cyt *c* (in buffer) can be seen in Figure 2A, and has comparable signatures reported earlier around 222 and 208 nm, which are characteristics of an all α-protein [54,74,75]. EG alone increases the CD signals, [*θ*]_222_ of the protein at high concentration, which resulted in a small increase in α-helical content and PEG 400 showed unaffected CD spectra of the protein, and hence no change in its secondary structure (see Figure 2A,B). Moreover, it can be observed from the results that in a mixture of crowders (PEG 400 + EG) of various concentration ratios leads insignificant increase in [*θ*]_222_ and α-helical content (overall showed no change). Besides structural studies, the far-UV CD can be used to analyze interaction of ligands with the protein, which shows structural loss [76,77,78]. The mixture of the crowders has a positive influence, leading to structural maintenance of the protein, and is more excluded than showing soft interactions as PEGs do alone [18,25,32,79]. Figure 2B depicted percentage values of secondary structure content (α-helical) of the protein in buffer, in PEG 400 and EG alone, and their mixtures, which were estimated from the values of [*θ*]_222_, using an equation of Morrisett et al. (1973) [46]. The values are provided in Table 1. The full and comprehensive study of binding sites will necessitate the use of a wide range of big-data approaches, including sequence-based bioinformatics, structural bioinformatics, computational chemistry, and molecular physics [80]. To assess binding sites and the binding occurrences within them, it is, therefore, necessary to combine both theoretical and experimental methods [81].

Protein-polymer solutions mimic the crowded environment in the cell [10,81,82,83,84,85,86,87]. Biomolecules present inside living cells execute their functions not only when exposed to only a single macromolecule, but are comparatively bared to different macromolecules with unusual and various types of shapes and sizes [24,88,89]. There are so many reports where protein folding has been studied under physiological conditions exposed to crowders, such asPEGs, dextran, ficoll, and natural macromolecules [20,22,79,88,89,90,91,92,93,94,95]. Our study showed two different molecules (EG and PEG 400) alone had contrary effects on the protein’s tertiary structure; EG stabilizes and PEG 400 perturbs the tertiary structure of the protein under similar conditions. However, it was observed that when PEG 400 and EG were combined, EG counteracts the perturbation of PEG 400. EG acts as kosmotrope (results compaction of protein via preferential exclusion or other interactions between co-solvents) as previously reported [37,49], while PEG 400 interacts with various proteins through chemical interactions and causes structural disruptions [22,30,31,32]. However, together they may form a complex (PEG 400-EG complex) which leads to the exclusion by the protein or unusual and unspecific binding on the protein, results decreases in the perturbation of the protein [18]. Our recent study where PEG 4000 was used as a crowder showed structural perturbation of cyt *c* without significant change in the heme-protein interaction at higher concentrations [96]. The heme protein in the crowded condition studied earlier also has shown retention of heme [94]. Crowding seems to be another method of mechanical super-competition that is devoid of certain markers that could assist in the detection and control of certain diseases, according to the experimental findings reported in the study [96]. Moreover, it was suggested that crowders at specific concentrations interact via soft interactions with proteins may be used to develop as therapy for various diseases [96].

In addition, in silico approaches are widely used for theoretical studies of proteins and other macromolecules, as well as changes that occur in the presence of various physiological and molecular events including mutations, ligand binding, temperature or pH changes, and the effects of different solutes/solvents (stabilizing and destabilizing agents) [57,60,97]. A quantitative theory of protein folding can give an accurate prediction using hypothetical models and simulations performed under variables that closely resemble those used in experiments (in vitro studies) [97].

MD simulations studies for the dynamics of the folding of cyt *c* in the presence of water, PEG 400, EG, mixtures of PEG 400 and EG, and 6 M GdmCl shown in Figure 3 provide changes in the parameters, *R*_g_, RMSD, RMSF and SASA of the protein. Table 2 provides the change in the values of these properties of the protein in the absence and presence of EG, PEG 400, mixtures of PEG 400 and EG, and 6 M GdmCl.

Values *R*_g_, RMSD, RMSF and SASA of the protein showed insignificant change in the presence of EG (50 and 300 mg/mL), significant decrease in the presence of PEG 400 (300 mg/mL), however an increase occurs in the mixture of crowders (PEG 400 and EG), and 6 M GdmCl shows maximum change in all parameters (Table 2). *R*_g_ is a probe to study the opening of protein conformation as it indicates the axial scattering of the atoms of the center of mass of the protein [64]. In addition to the *R*_g_, RMSD can be used to determine structural changes in the protein structure as a result of solvent effects [98]. We, therefore, estimated RMSD values of the α-carbon atoms for all the systems using the MD simulation trajectories acquired from the 100 ns simulation runs (see Figure 3 and Table 2). It is seen from Figure 3 that in the native conformation (black), the value which goes up around 40 nanoseconds (ns) simulation time and reaches convergence. Similarly, the protein in 6 M of GdmCl solution also shows a gradual elevation in the parameters which escalates untilthe end of the simulation time. In the presence of EG there are no significant changes (i.e., comparable to the native state), however fluctuation is more in the presence of PEG 400 alone (similar to the denatured one, 6 M GdmCl). Moreover, mixture of the crowders (PEG 400 + EG) causes decrease in the values of *R*_g_ (hence compaction of the protein compared to that of PEG 400 alone), SASA, RMSD and RMSF initially increase, in the middle (RMSF only when EG is greater in concentration (50 mg/mL) to that of PEG 400 (300 mg/mL) and insignificant increase (comparable to the protein in the presence of PEG 400 alone) and comparable with that of native conformation at end of simulation. From Figure 4 and Table 2 it can be confirmed that EG alone has an insignificant effect on the protein, PEG 400 alone and 6 M GdmCl leads structural disturbance, and mixtures of crowders (PEG 40 and EG) leads protein towards folded condition [18].

Due to the apparent noise in Figure 3 which is due to the high number (7) of plots superimposed for the sake of comparison. For the better understanding of the readers the Appendix A has been provided in the form of Lowess Curve Plots for *R*_g_, RMSD, and SASA as Appendix A. The Lowess plots are helpful in observing the trends on line plots having higher number of variables plotted as function of time.

Figure 4 showed secondary structural changes in cyt *c* in the presence of (Figure 4A) water, (Figure 4B,C) EG (50 and 300 mg/mL), (Figure 4D) PEG 400 (300 mg/mL), (Figure 4E,F) mixtures (PEG 400 + EG), and (Figure 4G) high concentrations of GdmCl (6 M). Analysis of the data using the DSSP program incorporated with GROMACS to identify secondary structure element changes during MD simulations. Results showed that the secondary structure is not influenced by any of the crowders (increases the structure of the protein), however the protein was observed also not affected in the presence of a mixture of crowders, which maintains the structure of the protein. Moreover, the MD simulation of protein in the presence of 6 M GdmCl (Figure 4G) showed a depth at initial time (15–30 s), however the secondary structure was not affected by GdmCl either. The overall studies showed that the secondary structure of the protein was not affected in the presence of PEG 400, EG and the mixture of crowders. Moreover, it can be observed from all the results that the mixture of crowders leads to a decrease in the perturbation of tertiary structure.

The mixtures of ethylene glycol with water are a prominent example of media with variable viscosity [35]. Water and ethylene glycol (EG) are perfectly miscible, and the viscosity of EG is much larger than that of pure water. Therefore, EG-water mixtures are a prominent example of media with controllable viscosity. This makes them useful for investigations of solvent dynamics effects on electron transfer reactions (i.e., the dependence of the mechanism of elementary act on solvent viscosity, saddle point avoidance) [35]. Moreover, molecular docking studies (Figure 5 and Figure 6) confirmed that PEG 400 and EG interact with cyt *c* via non-covalent forces without interacting with heme. PEG 400 interacts with some of the amino acids of the protein such as Arg38, His33 and Gly24 via single bonds of bond distance 3.26, 2.95 and 3.29 Å, respectively, and with Asn31 via more than two bonds with the distance of 2.81, 3.01 and 2.93 Å. Also, EG interacts with His33 via two bonds with distances of 3.01 and 2.98 Å and Asn31 via a single bond of distance 2.98 Å. 2D-structure model provides more information that PEG 400 and EG interact with the amino acid residues of the protein via non-covalent interactions that include conventional H-bonding, vander Waals forces, unfavorable accepter-accepter, and unfavorable donor-donor bonds. By computational analysis, it was determined that binding energy was −3.3 kcal mol^−1^ for PEG 400 and −2.7 kcal mol^−1^ for EG on interaction with cyt *c*. The molecular docking of these crowders (EG and PEG 400) individually with cyt *c* has been carried out earlier [32,37], where PEG 400 showed good binding energy (−3.5 kcal mol^−1^) as compared to EG (−2.9 kcal mol^−1^) and both showed a difference of 0.2 kcal mol^−1^ with that of additive effect. The understanding of intermolecular interactions that drive biological processes, structural biology, and structure-function correlations is aided by binding affinity (*K*_d_) [99]. Intermolecular weak contacts usually non-covalent interactions such as electrostatic interactions, van der Waals forces, hydrogen bonding and hydrophobic interactions, influence binding affinity. The binding affinity of crowders (PEG 400 and EG) was calculated by the equation: (Δ*G* = −*RT*ln*K*_b_) [100]. The values for Δ*G*, change in binding free energy is calculated from docking studies for crowders PEG 400 and EG on interacting with cyt *c* which are −3.3 and −2.7 kcalmol^−1^, respectively, using values of 1.987 cal mol^−1^ K^−1^ for *R* (gas constant) and 298 K for *T* (absolute temperature). The value of *K*_b_ (binding constant) was obtained from the equation, which in turn was used to calculate the binding affinity (*K*_d_) which was less for EG and greater for PEG 400 (Table 3). This table provides the detailed information of specific amino acids of the protein interacting with crowders (EG and PEG 400), hence binding sites on the protein and different possible interactions occurring.

The binding sites of EG and PEG 400 on Mb were found to be different, as were the residues involved in the interaction [18]. However, it can be seen in Figure 4 and Figure 5 and Table 3 that most of the amino acids of cyt *c* are similar which participates in the interaction with EG and PEG. In other words, observations and analysis from molecular docking studies showed that the binding site for PEG 400 and EG are identical in the protein (cyt *c*), however, the observed binding affinity is not the same. EG showed less binding as compared to PEG with cyt *c* can be seen in Table 3.

The general conclusion is that two systems operate simultaneously in medium (in vitro) to monitor changes in protein structure [20,101] which maintains the structure of the protein in crowded conditions. Shahid et al. demonstrated that crowders in mixtures stabilize lysozyme with a decrease in activity as crowder concentration was increased; this reflected the “stability-activity trade-off” theory [88]. Additionally, they confirmed that macromolecular crowding in mixtures has a more stabilizing effect compared to individual crowder molecules [88]. Such protein investigations in crowded conditions have made great progress from in vitro to truly imitating intracellular settings, i.e., in vivo. The combined crowding impact is more stabilizing than the sum of the two individual crowder molecules [25,88,102]. We recently reported that excluding volume (repulsive forces) stabilizes proteins, whereas soft interactions (attractive forces) destabilize them [22,32]. From this study, and recently published studies in the mixture of crowders [18,24,25,102], we suggested that both forces are necessary to maintain the protein’s structure and function and that in cellular conditions, these forces work together to maintain protein-protein interactions and protein-macromolecule interactions (i.e., stabilization-destabilization to maintain equilibrium), which in turn maintain cellular activities [7,103,104]. Molecules present in the system exclude one another effectively and these result in repulsion and hence avoid overlapping, therefore this is termed the hard-core or van der Waals repulsion. The researchers had shown that the addition of large size crowder molecules leads to an increase in the activity of the protein (SH3 in *Escherichia coli* cells) because its excluded volume is not accessible to the crowder molecules in the center [105]. Because the folded state of the protein has a smaller excluded volume, the change in activity caused by the crowder molecules promotes the native state of the protein [105]. Furthermore, the study showed that when the excluded volume of the crowding molecule increases, this effect should increase [105,106]. The reports have suggested that mixed crowding induces greater co-operativity in the domain movement as compared to the components of the mixtures. When compared with those obtained for individual crowders experimental investigations showed significant deviations from ideality, with an ideal solution being considered to be that arising from the sum of the contributions of those obtained in the presence of individual crowding agents [24].

If crowding by macromolecules in the cell’s solvent volume is not present, the volume may be occupied by other tiny molecules, primarily by water and ions that can overlap each other and also influence proteins in the milieu. Therefore a single large crowding macromolecule, for example, a protein or another biomolecule, will dislocate many of the smaller solvent molecules which help in the reduction inthe exclusion effect on protein activity [105,106]. Other interactions (non-specific, soft and attractive interactions) between the crowder and the test protein tend to cancel out hard-core stabilizing effect hence resulting protein instability [20,96,101]. Crowding effects can also be seen in the protein-metabolite interactions. Crowding and its impact on mechanisms and structural alterations have been proposed by studies of substrate binding and product release during enzyme reactions under crowded conditions [107]. Cyt *c* performs various functional roles in cellular conditions and must be influenced due to crowding as observed under in vitro conditions [32,37,96]. Due to weak interactions, cellular crowding surely increases the chances of biomolecules hitting repeatedly [108]. Various non-covalent interactions (weak interactions as well as strong interactions) have an important role in protein folding both in vitro and in the cellular conditions [20,105,108].

## 5. Conclusions

Proteins mediate many actions in the cell, including self-assembly of larger complexes, recognition and signaling; hence, the effect of crowding on proteins is significant, and their interactions with diverse macromolecules (including proteins) are of great interest. This study showed that crowders alone and crowders in the mixture behave differently with the protein (cyt *c*). Our overall observations (from in vitro and in silico) confirmed that cyt *c* in the presence of EG alone and in the system (PEG 400+ EG) either stabilizes the protein structure (both tertiary and secondary structure) or directs denatured fractions towards native form. Conversely, PEG alone leads to perturbation of the tertiary structure without influencing the secondary structure of the protein (confirmed from both in vitro and in silico methods). Molecular docking studies showed that both EG and PEG 400 interacts with cyt *c* via weak forces of attraction, which influences the protein accordingly. It was also observed that both of the crowders have identical binding sites on the protein with different free energies and binding affinity. Moreover, it can be said that EG reduces the impact of PEG 400 on protein structure and stability, favoring the exclusion volume effect over soft interactions in the crowded systems and allowing refolding of the PEG 400-induced denaturation. Such outcomes were already observed from our study where myoglobin (Mb) was exposed to similar conditions [18]. Therefore, it may be essential to include attractive interactions based on the hydrophobicity of crowding agents and protein residues, to represent more accurately solvent-mediated interactions occurring between crowders and proteins in cellular conditions by mimicking the observations under in vitro systems.

## Figures and Tables

**Figure 1 polymers-14-04808-f001:**
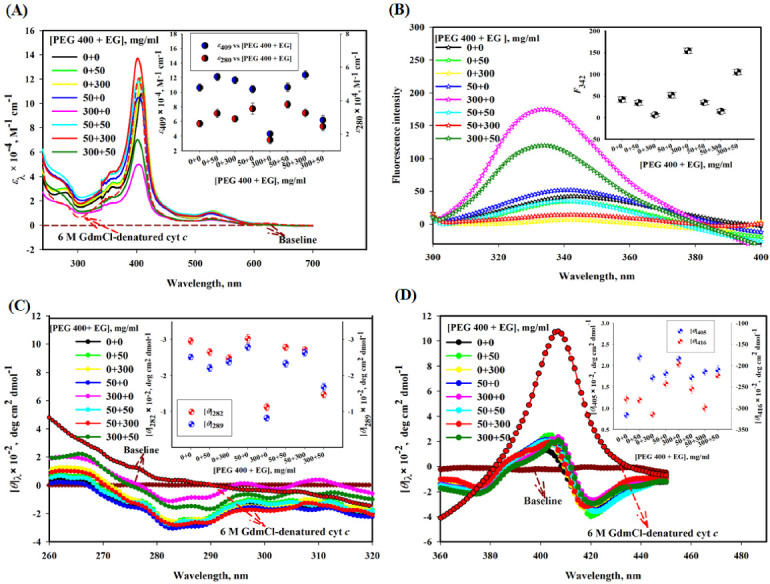
(**A**) Absorption, (**B**) tryptophan fluorescence, (**C**) near-UV CD and (**D**) Soret-CD spectra of cyt *c* with different [PEG 400 + EG] at pH 7.0 and 25 °C. Inset in each figure displays a plot of *ԑ*_409_ and *ԑ*_280_, *F*_342_, [*θ*]_282_ and [*θ*]_289_, [*θ*]_405_ and [*θ*]_416_, respectively, versus [PEG 400 + EG].

**Figure 2 polymers-14-04808-f002:**
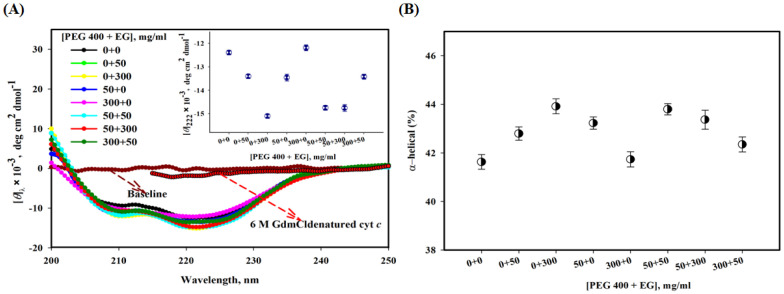
(**A**) Far-UV CD spectra of cyt *c* with different [PEG 400 + EG], and 6 M GdmCl at pH 7.0 and 25 °C.The inset of the figure displays a plot of [*θ*]_222_ versus [PEG 400 + EG]. (**B**) α-helical content of cyt *c* in the presence of mixtures of crowders.

**Figure 3 polymers-14-04808-f003:**
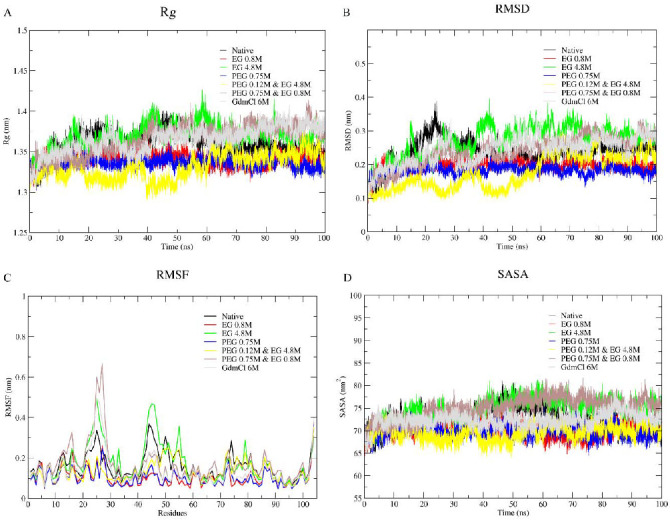
Line plots showing *R*_g_ values (**A**)**,** RMSD of Cα atoms (**B**), RMSF (**C**), and SASA (**D**) of cyt *c* in water and in various solutions.

**Figure 4 polymers-14-04808-f004:**
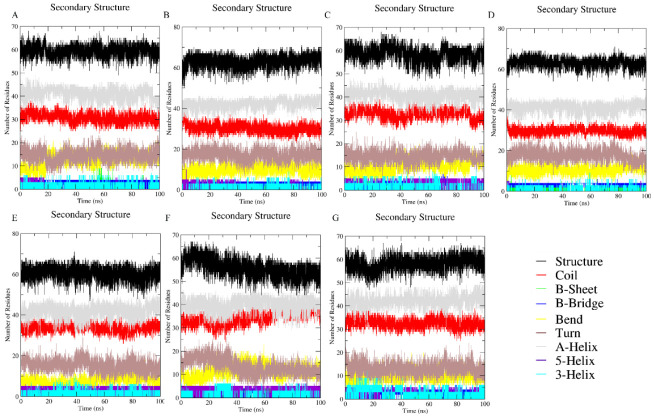
Fluctuations in the structure elements of cyt *c* in (**A**) water, (**B**) 50 mg/mL (0.8M) of EG, (**C**) 300 mg/mL (4.8 M) of EG, (**D**) 300 mg/mL (0.75 M) of PEG 400, (**E**) 300 mg/mL of PEG 400 (0.75 M) + 50 mg/mL of EG (0.8 M), (**F**) 50 mg/mLof PEG 400 (0.125 M) + 300 mg/mL of EG (4.8 M), and (**G**) GdmCl (6M) solutions.

**Figure 5 polymers-14-04808-f005:**
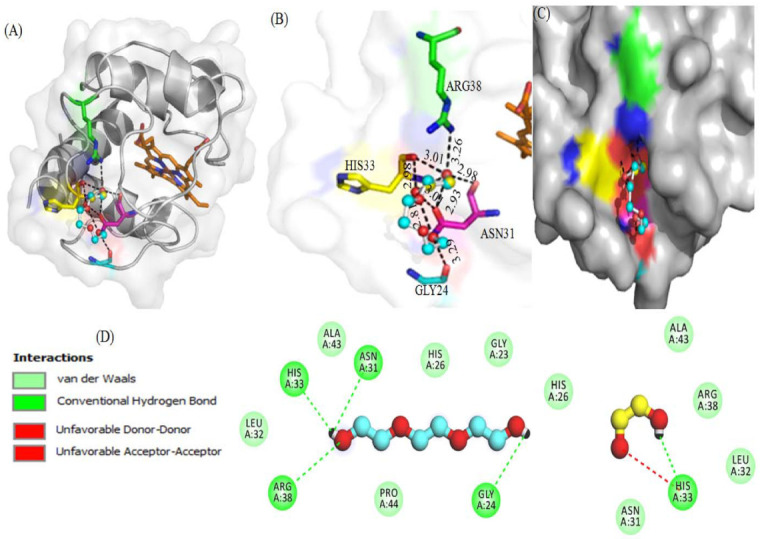
(**A**,**B**) represent interactions of crowders (i) ball and stick model of PEG 400 (blue) and (ii) EG (red-pink) with amino acid residues of cyt *c* (cartoon model, gray). (**C**) Surface view of cyt *c* and the binding site for PEG 400 and EG on the protein. (**D**) 2D-representation of various types of interactions between amino acid residues of the protein and crowders (PEG 400 and EG).

**Figure 6 polymers-14-04808-f006:**
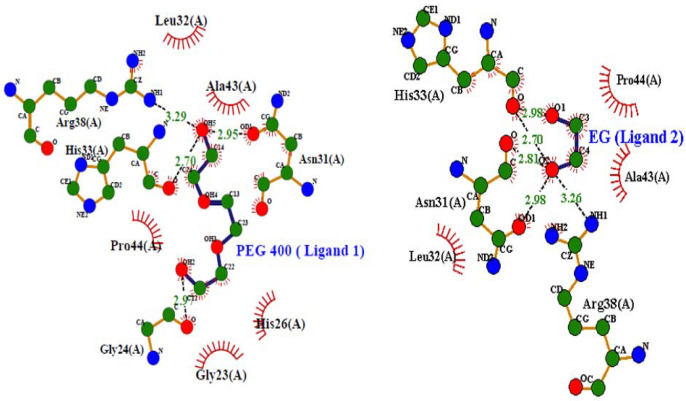
LigPlots showing the various types of interactions of PEG 400 and EG with residues of cyt *c*.

**Table 1 polymers-14-04808-t001:** Secondary structural content (%) of cyt *c* in the presence of various crowded mixture systems (PEG 400 + EG) measured at 222 nm.

[PEG 400 + EG], mg/mL	^a^ MRE at 222 nm, [*θ*]_222_	^b^ % α-Helical Content
0 + 0	−12,488 (±530)	41.6 (±1.2)
0 + 50	−13,306 (±632)	42.5 (±1.4)
0 + 300	−15,199 (±690)	44.1 (±1.2)
50 + 0	−13,599 (±566)	43.2 (±1.3)
300 + 0	−12,199 (±532)	41.4 (±1.1)
50 + 50	−14,633 (±663)	43.8 (±1.4)
50 + 300	−14,603 (±704)	43.3 (±1.1)
300 + 50	−13,522 (±643)	42.0 (±1.4)

^a^ Mean residual ellipticity; A ± with each parameter signifies the mean error; ^b^ Calculated by using Morrisett et al. (1973) equations [46].

**Table 2 polymers-14-04808-t002:** Average values of *R*_g_, RMSD, RMSF, and SASA obtained from MD simulations of cyt *c* in different solvent conditions.

Cyt *c* (Different Solvent Conditions)	*R*_g_ (nm)	RMSD (nm)	RMSF (nm)	SASA (nm^2^)
Water	1.359 (±0.014)	0.237 (±0.037)	0.162 (±0.068)	72.73 (±2.280)
50 mg/mL (0.8 M) EG	1.34 (±0.008)	0.207 (±0.017)	0.120 (±0.051)	70.17 (±1.59)
300 mg/mL (4.8 M) EG	1.37 (±0.015)	0.240 (±0.046)	0.174 (±0.096)	74.64 (±2.49)
300 mg/mL (0.75 M) PEG 400	1.30 (±0.007)	0.182 (±0.014)	0.105 (±0.046)	69.89 (±1.415)
300 + 50 mg/mL (0.75 + 0.8 M) PEG 400 + EG	1.341 (±0.0016)	0.190 (±0.051)	0.135 (±0.053)	70.59 (±1.68)
50 + 300 mg/mL (0.125 + 4.8 M) PEG 400 + EG	1.365 (±0.0018)	0.240 (±0.044)	0.172 (±0.081)	74.9 (±2.62)
GdmCl (6 M)	1.269 (±0.0016)	0.167 (±0.041)	0.098 (±0.081)	88.152 (±1.623)

**Table 3 polymers-14-04808-t003:** Binding sites and various types of interactions occurring between crowders (EG and PEG 400) and cyt *c*.

Crowder Molecule	Bonds	Amino Acid Interacted	Type of Interactions	Bond Distance (Å)	^b^ Δ*G* (kcal mol^−1^)	^c^*K*_b_ (M^−1^)
EG	1	Asn31 ^a^ (additional amino acids: Leu 32 ^a^, Arg38 ^a^, Ala43 ^a^, His 26 ^a^, Pro44 ^a^)	Vander Walls and hydrophobic interactions	2.98	−2.7	0.99 × 10^2^
	2	His33 ^a^	Conventional hydrogen bond and unfavorable donar-donar or acceptor-acceptor bond	3.01 and 2.98		
PEG 400	1	Arg38 ^a^, His33 ^a^ Gly24 (Additional amino acids: Leu 32 ^a^, Arg38 ^a^, Ala43 ^a^, His26 ^a^, Gly23, Pro44 ^a^)	Conventional hydrogen bond Conventional hydrogen bond Conventional hydrogen bond Vander Walls and hydrophobic interactions	3.26 2.95 3.29	−3.3	2.6 × 10^2^
	3	Asn31	Conventional hydrogen bonds	2.81, 3.01 and 2.93		

**^a^** Interacts with both EG and PEG 400; ^b^ Binding energy, ^c^ Binding affinity = 1/*K*_a_ (1/binding constant).

## Data Availability

Not applicable.

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
