# Peer review of "Measuring Structural Changes in Cytochrome c under Crowded Conditions Using In Vitro and In Silico Approaches"

_polymers, 2022, doi:10.3390/polym14224808_

Round 1

Reviewer 1 Report

Parray et al. explored the effect of combining crowding agents in the structure of a model protein: cytochrome C. The thematic is relevant, and the questions are interesting indeed. However, multiple serious flaws in the data analyses strongly compromise the conclusions, so I suggest rejecting the manuscript in this current format. Here are the major points and suggestions for the authors:

1) For all the data shown in figure 1, although the authors say that the differences are significant in the main text, the data has no error bars, and the overall changes are far from dramatic. Are these experiments done in triplicate? To be "significant" has a strong scientific meaning, based on rigorous statistical analysis. These analyses are not shown in the paper. For instance, the variations observed by the authors in the CD data (1C and 1D) are commonly observed for the same sample in a conventional bench CD if different spectra are not averaged automatically by the software. If the authors disagree, they should present the statistical data from replicate analyses with different sample preparations. 

2) Based on the number presented in table 1, it cannot be concluded that there are any significant changes in the α-helical content. Besides, the errors are not correctly propagated. E.g. (43.20 +- 2.6) should be (43.2 +- 2.6).

3) The redshift observed in figure 1B for the high PEG-containing conditions can also be because of increased protein compaction or dehydration in the Trp pocket regions. Both conclusions agreed with the other results in the present form. Besides, the fluorescence intensity cannot be negative.

4) The authors did not account for an effect caused by cumulative ionic, viscosity and osmotic effects of the polymers in the optical properties of the protein. If they believe that the effects are "crowding-agent"-specific, they should match as much as possible the overall physical-chemical properties in each comparison.   

4) There is no error propagation in the MD data shown in table 3. Because the MD simulations are done in a short timescale (100 ns), this is far from the required for secondary structure unfolding and refolding (for example, 10.1073/pnas.0308172101 and the references cited). Therefore, it should not be possible to observe significant secondary structure changes in this type of simulation. That is why the numbers in 6 M GdmCl (Fig. 4G) are pretty much the same as for all the other conditions, even though the authors did not include it in table 3.

5) General molecular docking (nearly) always gives positive hints in the analyses. For example, EG interacts with one residue (H33); therefore, this is unlikely to be specific, which is strongly suggested by the very low Kd observed (~10 mM). The authors should at least run a short MD simulation to support the observed interactions.

Minors:

1) figure 1 is heavily crowded, and the arrows are superposed with the texts difficulting the reading. The GA would also benefit from a more "clean" presentation.

Author Response

Comment:

Parray et al. explored the effect of combining crowding agents in the structure of a model protein: cytochrome c. The thematic is relevant, and the questions are interesting indeed. However, multiple serious flaws in the data analyses strongly compromise the conclusions, so I suggest rejecting the manuscript in this current format. Here are the major points and suggestions for the authors:

Response: The revised manuscript has been updated, the flaws in the data analysis has been removed following the suggestions of Reviewers.     

Comment 1:

 For all the data shown in figure 1, although the authors say that the differences are significant in the main text, the data has no error bars, and the overall changes are far from dramatic. Are these experiments done in triplicate? To be "significant" has a strong scientific meaning, based on rigorous statistical analysis. These analyses are not shown in the paper. For instance, the variations observed by the authors in the CD data (1C and 1D) are commonly observed for the same sample in a conventional bench CD if different spectra are not averaged automatically by the software. If the authors disagree, they should present the statistical data from replicate analyses with different sample preparations.

Response 1: The experimental measurements were carried out in triplicate. The statistical data has been presented in the revised manuscript in the replicate manner (with error bars) as per suggestion of the Reviewer.

Comment 2:

2) Based on the number presented in table 1, it cannot be concluded that there are any significant changes in the α-helical content. Besides, the errors are not correctly propagated. E.g. (43.20 +- 2.6) should be (43.2 +- 2.6).

Response 2: The data has been updated in the revised mnauscript as there were some errors on the basis of calculations. We are thankful to the Reviewer for providing the suggestion.

Comment 3:

The red shift observed in figure 1B for the high PEG-containing conditions can also be because of increased protein compaction or dehydration in the Trp pocket regions. Both conclusions agreed with the other results in the present form. Besides, the fluorescence intensity cannot be negative.

Response 3: It has been observed in the data published [1] that protein (cyt c) I the presence of Ethylene glycol (EG) leads quenching which increases to maximum at high concentration of EG.  In that condition the heme-Trp distance decreases results decrease in the fluorescence intensity [1]. At maximum wavelength the fluorescence (F342) is not zero it has positive value around (obtained from data 6.418) (Please see Figure 1a).

Figure 1a

Comment 4:

The authors did not account for an effect caused by cumulative ionic, viscosity and osmotic effects of the polymers in the optical properties of the protein. If they believe that the effects are "crowding-agent"-specific, they should match as much as possible the overall physical-chemical properties in each comparison.  

Response 4: As suggested by the Reviewer, the physical-chemical properties in each comparison of the crowder molecules have been included in the revised manuscript.

Comment 5:

 There is no error propagation in the MD data shown in table 3. Because the MD simulations are done in a short timescale (100 ns), this is far from the required for secondary structure unfolding and refolding (for example, 10.1073/pnas.0308172101 and the references cited). Therefore, it should not be possible to observe significant secondary structure changes in this type of simulation. That is why the numbers in 6 M GdmCl (Fig. 4G) are pretty much the same as for all the other conditions, even though the authors did not include it in table 3.

Response 5:  The MD simulation of protein in the presence of 6 M GdmCl (Figure 4G) shows a depth at initial time (15-30 s), which suggests loss of some amino acids and hence the structure perturbation, results decrease in the the alpha-helical content and increase in the values of coils, turns and bends (see Table 3). We have provided the values as such in the Table as obtained from the MD simulations (using DSSP software) and with the reason. 

Comment 6:

General molecular docking (nearly) always gives positive hints in the analyses. For example, EG interacts with one residue (H33); therefore, this is unlikely to be specific, which is strongly suggested by the very low Kd observed (~10 mM). The authors should at least run a short MD simulation to support the observed interactions.

Response 6: In the support of the data the MD simulation of the protein in the presence of EG has been carried out (see Figure 3).

Minors:

Comment 1:

Figure 1 is heavily crowded, and the arrows are superposed with the texts difficulty in the reading. The GA would also benefit from a more "clean" presentation.

Response 1:  We are thankful to the Reviewer for the suggestion; the GA has been updated in the revised manuscript.

Reviewer 2 Report

The authors described the changes of protein structures upon interacting with crowders, including ethylene glycol monomer and poly (ethylene glycol), from in vitro studies and in silico simulations. The research is interesting and of great importance in biology community. In general, this manuscript is well-organized and presented in a very logical manner. I recommend it for publication in Polymers after addressing the following concerns:

1). PEG 400 roughly has 7 EG repeating units. Why the authors choose PEG 400, instead of other higher molecular weight PEG polymers?

2). The figures of cyt c structures change in different solutions are not clearly presented. Figure 3 and Figure 4, it is hard to distinguish one signal from the other. The noise levels are high. Is it possible to use inserts or enlarged graphs to represent the data?

Author Response

Comments and Suggestions for Authors

Comment: The authors described the changes of protein structures upon interacting with crowders, including ethylene glycol monomer and poly (ethylene glycol), from in vitro studies and in silico simulations. The research is interesting and of great importance in biology community. In general, this manuscript is well-organized and presented in a very logical manner. I recommend it for publication in Polymers after addressing the following concerns:

Response: We are thankful to the Reviewer finding this piece of research work interesting and recommend it for the publication. We are addressing the suggestions and modify the revised manuscript in sight of comments from the Reviewer.

Comment 1:

1). PEG 400 roughly has 7 EG repeating units. Why the authors choose PEG 400, instead of other higher molecular weight PEG polymers?

Response 1: This is really a good question and well expecting. Actually the higher molecular weight sizes are more viscous which result various issues including the sample volume makeup, bubble formation and hence affects the data. Moreover, the nature and the type of deviation occurs much on the basis of length scale of the crowder which also depend on the identities of the components of the crowder mixtures. PEG 400 and EG both are in liquid form and we have already observed PEG 400 and EG alone effects on heme proteins, where PEG 400 has drastic changes on protein structures however EG has opposite effects [1-5]. Therefore it was better to choose such contrary outcomes together to see the variation.

Comment 2:

The figures of cyt c structures change in different solutions are not clearly presented. Figure 3 and Figure 4, it is hard to distinguish one signal from the other. The noise levels are high. Is it possible to use inserts or enlarged graphs to represent the data?

Response 2:

The apparent noise in Figure 3 is due to the high number (7) of plots superimposed for the sake of comparison. Keeping separate plots for comparing the effect of solvent on cyt c in comparison to the native structure would result in higher number of figures in the main text. Therefore, we presented the figures as they are in the manuscript. However, as per the suggestion and for the better understanding of the readers we have also provided the Lowess Curve Plots for Rg, RMSD, and SASA as supplementary figures (Figure S1). The lowess plots are helpful in observing the trends on line plots having higher number of variables plotted as function of time. We hope that we are able to fulfill the requirement of the reviewer.

Figure S1: The Lowess Curve Plots for Rg, RMSD, and SASA for the Figure 3 in the manuscript will be added in supplementary.

Reviewer 3 Report

I reviewed the following manuscript: Title: Measuring structural changes in cytochrome c under crowded conditions using in vitro and in silico approaches (Journal: Polymers, Manuscript ID: polymers-1717292). This paper describes the effect of molecular crowding agents on the structural change of cytochrome c. For this study purpose, the spectral measuring and calculations were performed. The important conclusion is the difference effects of PEG, EG, and their mixture. The reported contents are valuable for in vitro study using molecular crowding agents. However, the several points should be improved before publication. The noticed points are listed.

1) Table 3; How is the accuracy of the presented values? The difference of the presented ratio (dependence of solvent environments) should be explained quantitatively. Standard error or standard deviations should improve the results.

2) Discussion, line 388-389; The mechanism of protein structure change by polar environment should be explained. This information is helpful for readers.

3) Discussion or Conclusion; An image figure to explain the Table 4 and the conclusion may be helpful for readers to understand the effects of PEG and EG.

4) For practical reason, a protocol or presentation of optimal condition of the crowding agents should improve this paper. For example, the effective concentration and mixture ratio of PEG-400 and EG for experimental investigation of protein in vitro are important information to study protein (cytochrome c) under similar environment in cells.

5) I would like to recommend the spell check of manuscript.

Author Response

 Comments and Suggestions for Authors

Comment:

I reviewed the following manuscript: Title: Measuring structural changes in cytochrome c under crowded conditions using in vitro and in silico approaches (Journal: Polymers, Manuscript ID: polymers-1717292). This paper describes the effect of molecular crowding agents on the structural change of cytochrome c. For this study purpose, the spectral measuring and calculations were performed. The important conclusion is the difference effects of PEG, EG, and their mixture. The reported contents are valuable for in vitro study using molecular crowding agents. However, the several points should be improved before publication. The noticed points are listed.

 Response: We are thankful to the Reviewer finding this piece of research work interesting and recommended for the publication. We have improved the revised manuscript with suggestions in sight of comments from the Reviewer.

Comment 1:

Table 3; How is the accuracy of the presented values? The difference of the presented ratio (dependence of solvent environments) should be explained quantitatively. Standard error or standard deviations should improve the results.

Response: Table 3 has been modified and standard deviations for the change in percentages of secondary structure elements have also been included in the table.

Comment 2:

 Discussion, line 388-389; The mechanism of protein structure change by polar environment should be explained. This information is helpful for readers.

Response 2: The line “The mechanism of protein structure change by polar environment should” has been explained in the revised manuscript.

Comment 3:

Discussion or Conclusion; An image figure to explain the Table 4 and the conclusion may be helpful for readers to understand the effects of PEG and EG.

Response 3: The Table showed the various types of interactions of PEG 400 and EG with the protein, showing binding sites of these crowders on the protein. No doubt the Reviewer has spectacle this right that it will be helpful for readers to understand the effects of crowders alone and in mixture.

Comment 4:

For practical reason, a protocol or presentation of optimal condition of the crowding agents should improve this paper. For example, the effective concentration and mixture ratio of PEG-400 and EG for experimental investigation of protein in vitro are important information to study protein (cytochrome c) under similar environment in cells.

Response 4: We are very thankful to the Reviewers for suggesting to appraise the in vitro studies in terms of in vivo studies. Our current and upcoming projects will surely work on this.

Comment 5:

I would like to recommend the spell check of manuscript.

Response 5: As suggested by the Reviewer the spell check in the revised manuscript has been done.

  1. Parray, Z.A., et al., Conformational changes in cytochrome c directed by ethylene glycol accompanying complex formation: Protein-solvent preferential interaction or/and kosmotropic effect. Spectrochimica Acta Part A: Molecular and Biomolecular Spectroscopy, 2020. 242: p. 118788.
  2. Parray, Z.A., et al., First evidence of formation of pre-molten globule state in myoglobin: A macromolecular crowding approach towards protein folding in vivo. International Journal of Biological Macromolecules, 2019. 126: p. 1288-1294.
  3. Parray, Z.A., et al., Formation of molten globule state in horse heart cytochrome c under physiological conditions: Importance of soft interactions and spectroscopic approach in crowded milieu. International Journal of Biological Macromolecules, 2020. 148: p. 192-200.
  4. Parray, Z.A., et al., Structural Refolding and Thermal Stability of Myoglobin in the Presence of Mixture of Crowders: Importance of Various Interactions for Protein Stabilization in Crowded Conditions. Molecules, 2021. 26(9): p. 2807.
  5. Parray, Z.A., et al., Effects of Ethylene Glycol on the Structure and Stability of Myoglobin Using Spectroscopic, Interaction, and In Silico Approaches: Monomer Is Different from Those of Its Polymers. ACS omega, 2020. 5(23): p. 13840-13850.

Round 2

Reviewer 1 Report

I have rejected the first version of this manuscript based on multiple problems that heavily compromised the main conclusions, and most of them persist in the current version. I will also argue that there is a clear issue with the final model if they still support the conclusions in the present form. In my analyses on the current version of the paper, no robust data suggests that both crowding agents interact with the protein. However, if they do, the numerous effect observed might not be because of the molecular crowding effect but solely because of unspecific binding.

- On comment 2:

The values presented in figure 2b do not correspond with the ones from table 1. The errors are not presented correctly in figure 2b. If we take the one with lower helical content (0 + 0) as an example, the value is 41.6 +- 1.5. That means that the model says that you have an uncertainty of the correct value to be between 40.1 and 43.1. Therefore, the data shown in table 1 do not show any significant changes in the helical content under the different conditions.

- On comment 3:

I argued against the red-shift effect, not changes in the fluorescence intensity. Therefore, the question remains. Quenching effects can arise in a collisional way without involving any specific interaction with the protein or inducing changes in protein structure. Therefore, when the authors said:

Lines 476-479 - “A decrease in the intensity of the band (Soret region) without any shift in the wavelength in the presence of PEG 400 is greatest at 300 mg ml-1, which reflects that protein structure is perturbed and heme environment changes towards the polar environment.”

They need to present where they took this conclusion. For instance, the fluorescence quenching experiments to probe trp accessibility using the water-soluble acrylamide leads to total quenching of fluorescence intensity without any changes in wavelength or protein structure. Since EG is quite small, a collisional quenching effect only is a robust hypothesis.

Besides, the fluorescence intensity is significantly negative over 360 nm, and the authors need to explain why.

- on comment 4

I could not find where the authors have inserted it.

- on comment 5

From the data presented in figure 4, it is clear that there is no significant variation in the overall structural content. A <5 ns scale fluctuation can be random in this type of MD simulation, especially because it is not very long. Besides, the authors inserted in table 3 the value for alfa helix as 31% for the GdmCl, but the number for all the other conditions are clearly not corresponding with figure 5. For this particular one, I guess they inserted the alfa-helix content (around 32% for all), but for all the others, they inserted the coil fraction (around 40% because it adds the other helix elements). Therefore, I still argue that “it should not be possible to observe significant secondary structure changes in this type of simulation”, especially in this time scale. The data here, just like the one presented in table 1, suggest that there are no significant changes in the alfa-helix content.

-on comment 6.

I was arguing about the EG molecule, not the protein. Because what is stabilizing the EG is just one single hydrogen bond, this should not be specific and is probably not time-stable even in a short MD simulation. The authors need to show that this is not the case if they want to use this data to support their conclusions since most of the data shown suggest that most of the fluorescence effects could be just collisional ones.

Author Response

#Point to Point Response to the Comments of Reviewers

#Reviewer 1:

Comment:

I have rejected the first version of this manuscript based on multiple problems that heavily compromised the main conclusions, and most of them persist in the current version. I will also argue that there is a clear issue with the final model if they still support the conclusions in the present form. In my analyses on the current version of the paper, no robust data suggests that both crowding agents interact with the protein. However, if they do, the numerous effects observed might not be because of the molecular crowding effect but solely because of unspecific binding.

Response:  PEG 400 was earlier studied thoroughly for the proteins (cytochrome c and myoglobin), where it was observed that PEG 400 interacts via soft interactions with the proteins [1, 2]. However in earlier studies, EG has showed no such effect on the structure of Mb and doesn’t show strong binding at 25 oC but interacts strongly with the protein at high temperatures, where the protein is in denatured condition [3]. In case of cyt c, EG showed kosmotropic effect and shows weak interactions with the protein (shown by ITC as well as using various spectroscopic techniques) [4]. EG showed quenching of Trp in cyt c (as Reviewer said this may be colloisional quenching than static or dynamic), however proven by time resolved fluorescence to be static on the basis of plot of average life time ratio (t0/t) against concentration of EG, showed straight line than linear [4].

May be in case of mixture of crowders (EG + PEG 400), the binding is unspecific as suggested by the Reviewer and we have introduced this statement in the Re-revised manuscript.

We have removed the quenching word; we focused on increase and decrease of Fluorescence intensity in the re-revised manuscript.

Comment 1:  

On comment 2:

The values presented in figure 2b do not correspond with the ones from table 1. The errors are not presented correctly in figure 2b. If we take the one with lower helical content (0 + 0) as an example, the value is 41.6 +- 1.5. That means that the model says that you have an uncertainty of the correct value to be between 40.1 and 43.1. Therefore, the data shown in table 1 do not show any significant changes in the helical content under the different conditions.

Response 1: We agree with the Reviewer. We have rewritten the sentences in the Re-revised manuscript that the protein in the presence of different molecules (PEG 400, EG and mixtures) do not show any significant changes in the helical content, overall change is insignificant.

Comment 2:  

- On comment 3:

I argued against the red-shift effect, not changes in the fluorescence intensity. Therefore, the question remains. Quenching effects can arise in a collisional way without involving any specific interaction with the protein or inducing changes in protein structure. Therefore, when the authors said:

Lines 476-479 - “A decrease in the intensity of the band (Soret region) without any shift in the wavelength in the presence of PEG 400 is greatest at 300 mg ml-1, which reflects that protein structure is perturbed and heme environment changes towards the polar environment.”

They need to present where they took this conclusion.

For instance, the fluorescence quenching experiments to probe trp accessibility using the water-soluble acrylamide leads to total quenching of fluorescence intensity without any changes in wavelength or protein structure. Since EG is quite small, a collisional quenching effect only is a robust hypothesis.

Besides, the fluorescence intensity is significantly negative over 360 nm, and the authors need to explain why.

Response 2: We have rewritten the sentence in the re-revised manuscript as suggested by the Reviewer.  

[Lines 476-479 - “A decrease in the intensity of the band (Soret region) without any shift in the wavelength in the presence of PEG 400 is greatest at 300 mg ml-1, which reflects that protein structure is perturbed and heme environment changes towards the polar environment.”].

The changes in the Soret band around 409 nm suggests the change in the heme-protein interactions, hence the vicinity around the heme, which confirms that perturbation of tertiary structure occurs in the protein, The reports have shown also the decrease in the absorbance around 409 nm in heme protein, showing protein perturbation [2, 5-7]. The citations have been inserted in the revised manuscript also.

The fluorescence intensity is significantly negative not over 360 nm, but above 380 nm. This occurred may be due to fluorescence of ligand and/or buffer intrusion, which was subtracted from the main spectra and give final spectra seen in Figure 1(B).

Comment 3:

- on comment 4

I could not find where the authors have inserted it.

Response 3: We have added the details in the Re-revised manuscript as suggested by the Reviewer.

Comment 4:

- on comment 5

From the data presented in figure 4, it is clear that there is no significant variation in the overall structural content. A <5 ns scale fluctuation can be random in this type of MD simulation, especially because it is not very long. Besides, the authors inserted in table 3 the value for alfa helix as 31% for the GdmCl, but the number for all the other conditions are clearly not corresponding with figure 5. For this particular one, I guess they inserted the alfa-helix content (around 32% for all), but for all the others, they inserted the coil fraction (around 40% because it adds the other helix elements). Therefore, I still argue that “it should not be possible to observe significant secondary structure changes in this type of simulation”, especially in this time scale. The data here, just like the one presented in table 1, suggest that there are no significant changes in the alfa-helix content.

Response: The values in the Table were obtained from the MD simulations (using DSSP software) and we decided to remove Table 3 in the re-revised manuscript for no more imperfections and queries on it.  As suggested by the reviewer we have rewritten the sentence in the re-revised manuscript that the change of secondary structures is insignificant in the presence of EG, PEG 400 and mixture of crowders as well as GdmCl.

Comment 4:

-on comment 6.

I was arguing about the EG molecule, not the protein. Because what is stabilizing the EG is just one single hydrogen bond, this should not be specific and is probably not time-stable even in a short MD simulation. The authors need to show that this is not the case if they want to use this data to support their conclusions since most of the data shown suggest that most of the fluorescence effects could be just collisional ones.

Response 4:

The mixtures of ethylene glycol with water are a prominent example of media with variable viscosity.

 As the Reviewer suggested running MD simulation of EG only in previous Comment 6, however our system on which we run MD simulations is not working due to some issues.

Though, the reports (Kaiser, A., et al. J. Phys. Chem. B 2016, 120, 10515−10523, DOI: 10.1021/acs.jpcb.6b05236) [8] had showed classical molecular dynamics simulations at room temperature for mixtures of ethylene glycol (EG) and water with EG mole fractions (XE). They found a slightly overestimated slowdown of the dynamics with increasing EG concentration compared to experimental data. Statistics of the hydrogen bond network and hydrogen bond lifetimes were derived from suitable time correlation functions and also show a slowdown in the dynamics with increasing xE. 

Water and ethylene glycol (EG) are perfectly miscible and the viscosity of EG is much larger than that of pure water. Therefore, EG−water mixtures are a prominent example of media with controllable viscosity. This makes them useful for investigations of solvent dynamics effects on electron transfer reactions (i.e., the dependence of the mechanism of elementary act on solvent viscosity, saddle point avoidance) [9].

  1. Parray, Z.A., et al., First evidence of formation of pre-molten globule state in myoglobin: A macromolecular crowding approach towards protein folding in vivo. International Journal of Biological Macromolecules, 2019. 126: p. 1288-1294.
  2. Parray, Z.A., et al., Formation of molten globule state in horse heart cytochrome c under physiological conditions: Importance of soft interactions and spectroscopic approach in crowded milieu. International Journal of Biological Macromolecules, 2020. 148: p. 192-200.
  3. Parray, Z.A., et al., Effects of Ethylene Glycol on the Structure and Stability of Myoglobin Using Spectroscopic, Interaction, and In Silico Approaches: Monomer Is Different from Those of Its Polymers. ACS omega, 2020. 5(23): p. 13840-13850.
  4. Parray, Z.A., et al., Conformational changes in cytochrome c directed by ethylene glycol accompanying complex formation: Protein-solvent preferential interaction or/and kosmotropic effect. Spectrochimica Acta Part A: Molecular and Biomolecular Spectroscopy, 2020. 242: p. 118788.
  5. Mondal, S. and B. Das, A study on the interaction of horse heart cytochrome c with some conventional and ionic liquid surfactants probed by ultraviolet-visible and fluorescence spectroscopic techniques. Spectrochim Acta A Mol Biomol Spectrosc, 2018. 198: p. 278-282.
  6. Ahmad, Z. and F. Ahmad, Mechanism of Denaturation of Cytochrome-C by Lithium-Salts. Indian Journal of Chemistry Section B-Organic Chemistry Including Medicinal Chemistry, 1992. 31(12): p. 874-879.
  7. Fisher, W.R., H. Taniuchi, and C.B. Anfinsen, On the role of heme in the formation of the structure of cytochrome c. J Biol Chem, 1973. 248(9): p. 3188-95.
  8. Kaiser, A., et al., Hydrogen Bonding and Dielectric Spectra of Ethylene Glycol–Water Mixtures from Molecular Dynamics Simulations. The Journal of Physical Chemistry B, 2016. 120(40): p. 10515-10523.
  9. Ismailova, O., et al., Interfacial Bond-Breaking Electron Transfer in Mixed Water–Ethylene Glycol Solutions: Reorganization Energy and Interplay between Different Solvent Modes. The Journal of Physical Chemistry B, 2013. 117(29): p. 8793-8801.

Reviewer 3 Report

I checked the revised manuscript.

Author Response

Response: We are thankful to the Reviewer for finding the revised manuscript suitable to be published in the Journal (Polymers MDPI) .
